# TNF-Related Apoptosis-Inducing Ligand: Non-Apoptotic Signalling

**DOI:** 10.3390/cells13060521

**Published:** 2024-03-16

**Authors:** Abderrahmane Guerrache, Olivier Micheau

**Affiliations:** 1Université de Bourgogne, 21000 Dijon, France; 2INSERM Research Center U1231, «Equipe DesCarTes», 21000 Dijon, France; 3Laboratoire d’Excellence LipSTIC, 21000 Dijon, France

**Keywords:** apoptosis, metastasis, migration, EMT, cancer, TNF, TRAIL, signalling

## Abstract

TNF-related apoptosis-inducing ligand (TRAIL or Apo2 or TNFSF10) belongs to the TNF superfamily. When bound to its agonistic receptors, TRAIL can induce apoptosis in tumour cells, while sparing healthy cells. Over the last three decades, this tumour selectivity has prompted many studies aiming at evaluating the anti-tumoral potential of TRAIL or its derivatives. Although most of these attempts have failed, so far, novel formulations are still being evaluated. However, emerging evidence indicates that TRAIL can also trigger a non-canonical signal transduction pathway that is likely to be detrimental for its use in oncology. Likewise, an increasing number of studies suggest that in some circumstances TRAIL can induce, via Death receptor 5 (DR5), tumour cell motility, potentially leading to and contributing to tumour metastasis. While the pro-apoptotic signal transduction machinery of TRAIL is well known from a mechanistic point of view, that of the non-canonical pathway is less understood. In this study, we the current state of knowledge of TRAIL non-canonical signalling.

## 1. Introduction

TNF-related apoptosis ligand (TRAIL) is known for its ability to trigger cell death, but increasing evidence indicates that TRAIL may also induce cell differentiation [1,2,3,4], tumour progression [5,6,7], invasion and metastasis [8,9,10,11,12,13,14]. These pleiotropic non-apoptotic signalling pathways are conveyed by TRAIL agonist cognate receptors, two receptors of the TNF receptor superfamily (TNFRSF) that share a large number of structural similarities (Figure 1), interacting partners and signalling pathways [15]. However, while most of these TNFRSF receptors are able to induce identical cellular outcome, such as cell death or cell motility, their *modus operandi* is not always similar, depending on both the receptor and ligand considered.

Likewise, although much less efficiently than TNFR1, both TRAIL and Fas-ligand agonist receptors are able to trigger NF-kB signalling, leading in some cases to increased tumour growth and inflammation [16,17,18,19,20,21,22,23,24,25,26,27,28,29,30] (Figure 2). Yet, with the exception of the molecular circuitry enabling engagement of cell death and maybe NF-κB, albeit orchestrated in a radically distinct manner in the case of TNF as compared to FasL or TRAIL (Figure 2), how TRAIL-non-apoptotic signalling complexes, including signal transduction leading to cell motility, are triggered at the molecular level remains ill understood.

This comprehensive review therefore aims at presenting the complexity of the TRAIL signal transduction machineries in light of the diversity of its roles or attributions to physiology and related pathologies, in order to provide a current understanding of the molecular mechanisms underlying non-apoptotic TRAIL signal transduction, with a focus on the molecular events leading to cell motility and metastasis.

## 2. The TRAIL System

### 2.1. TRAIL-Induced Cell Death

#### 2.1.1. TRAIL and Its Cognate Receptors

TRAIL was described for the first time as a pro-apoptotic ligand that induces apoptosis [31,32]. Encoded by the gene *TNFSF10*, in 3q26 position on human chromosome 3 [31,32], TRAIL is a type II transmembrane protein which has the ability to bind to six distinct receptors (Figure 1). Five of them are located within the same chromosome in position 8p21.3 for DR4 (TRAIL-R1 encoded by *TNFRSF10A* gene) [33], DR5 (two splice variants of TRAIL-R2 encoded by *TNFRSF10B* gene) [34,35,36,37,38], DcR1 (TRAIL-R3 encoded by *TNFRSF10C*) [39,40] and DcR2 (TRAIL-R4 encoded by *TNFRSF10D*) [41] and position 8q24.12 for OPG (osteoprotegerin encoded by *TNFRSF11B*) [42,43,44,45,46]. DcR3 is encoded by *TNFRSF6B* on chromosome 20 in position 20q13.33 [44].

Out of the six TRAIL cognate receptors, only DR4 and DR5 are able to trigger apoptosis because they harbour a death domain (DD) in their c-terminal part (Figure 1). The DD is also found in TNF-R1 and Fas [47,48,49,50,51,52], and is necessary and sufficient to engage the pro-apoptotic machinery [47,48]. The remaining receptors are unable to induce apoptosis due either to the absence of a functional DD, in their intracellular c-terminal portion, or because these receptors are secreted to the extracellular compartment [53] (Figure 1). DcR1 and DcR2 are expressed on the cell surface, thanks to a GPI-anchor, or a transmembrane protein, but are devoid of a functional DD and are thus also unable to trigger apoptosis. The two other antagonist receptors, OPG [54] and DcR3 [55] are secreted as soluble receptors in the extracellular compartment, and are thus unable to transduce cell death, including OPG which harbours two DD (Figure 2). Regardless of their subcellular localisation, all four antagonist receptors are capable of competing with TRAIL to inhibit apoptosis-induced by either DR4 or DR5 [42,43,56,57,58,59,60] (Figure 1 and Figure 2).

#### 2.1.2. Canonical TRAIL-Induced Apoptosis Signal Transduction

TRAIL binding to its two agonist receptors, DR4 and DR5, leads to the formation of homo or hetero multimeric complexes on the cell surface, which in turn enable the recruitment of the adaptor protein FADD (fas-associated via death domain) and the initiator pro-caspases-8 and/or -10, leading to the formation of the death-inducing signalling complex or DISC [20,61,62,63,64], in which the initiator caspase-8, similar to the Fas DISC, is activated by mere proximity-induced dimerization [65,66,67]. Once activated this initiator caspase, self cleaves itself, enabling it not only to free itself from the DISC, but also to reach and cleave substrates localized in the cytosol, such as the executioner caspases-3, -6 and -7 [68], which ultimately will concur in the execution of apoptosis, culminating in DNA fragmentation and the formation of apoptotic bodies [69]. 

In the late 1990’s, two type of cells, found to rely or not, on the activation of mitochondria, were described to transduce differentially apoptosis upon Fas ligand and TRAIL stimulation [70,71]. In type I cells, sufficient caspase-8 is activated to undergo apoptosis [72], regardless of mitochondria [73,74]. The intrinsic pathway is, however, required in type II cells, to fully transduce apoptosis upon TRAIL or FasL stimulation. Likewise, contrary to type I cells, mere loss of Bax expression [75] or overexpression of Bcl-2 anti-apoptotic members [72,76,77], is sufficient to abrogate the execution of apoptosis in type II cells. Activation of the mitochondrial pathway by TRAIL receptors is mediated, in these cells, by a caspase-8-dependent cleavage of Bid [71,78], a BH3-only Bcl-2 family member, whose cleavage allows truncated Bid (tBid) insertion into mitochondrial membranes where it induces the translocation and oligomerization of Bax and Bak [79,80,81], inducing the release of cytochrome-c (Cyt-c). Once released from the outer membranes of mitochondria, cytochrome c forms, together with the initiator caspase-9 and APAF-1 (Apoptotic peptidase activating factor-1), the apoptosome complex [82,83,84], which allows the activation of the caspase-9 by mere dimerization [85] and which culminates in the activation of the executioner caspases (Figure 3).

The outcome of this fine orchestration is also highly controlled by genetically regulated events, as well as cellular heterogeneity and stochasticity. Likewise, it has been demonstrated that random assembly of the receptors upon ligand stimulation [86], as well as intracellular or membrane-bound proteins stochastic distribution during cell division [87], may contribute to cell fate decision.

#### 2.1.3. TRAIL-Induced Necroptosis

Besides apoptosis, TRAIL-induced cell death may proceed through necroptosis, in specific cell types or under certain conditions. Like TNFα and FasL, TRAIL has been found, in a seminal work by the late Pr Jurg Tschopp [88], to induce necroptosis in a RIPK1-dependant manner, in the presence of a pan-caspase inhibitor or in the absence of FADD, in the human jurkat T-cell line [88]. It was next found that at acidic extracellular pH (pHe), a condition that can be encountered in the tumour microenvironment (TME), TRAIL-induced cell death proceeds through necroptosis. Likewise, mere acidification of the extracellular pH, in vitro, switches TRAIL-induced cell death from apoptosis to necroptosis [89]. This switch was also found to require RIPK1 [90]. 

The first inhibitor of this programmed inflammatory cell death, the necrostatine [91], was later found to inhibit RIPK1 [92]. RIPK1 is an integrator of cellular stimulation with protein kinase activity and scaffolding functions. It is composed of a N-terminal kinase domain, an intermediary domain (ID), a C-terminal homology interaction motif (RHIM), and a DD. Owing to homotypic interactions, RIPK1 can be recruited to DD-containing receptors through its DD, and provided that it is not cleaved by the caspase-8 within the DISC [93,94], RIPK1 can recruit RIPK3 through the RHIM [95,96] and phosphorylate RIPK3 [97,98,99,100], forming the ripoptosome [101], which then phosphorylates and activates the pseudo kinase mixed lineage kinase domain-like protein (MLKL) [102,103]. Activation of MLKL leads to its oligomerization, translocation to the plasma membrane, forming large pores which engage ion channels to mediates ion influx, cell swelling and plasma membrane rupture followed by the uncontrollable release of intracellular material [102,104,105,106] (Figure 4). 

Changes in pH occur naturally in the vicinity of tumour cells [107] as well as during ischemia [108]. The latter are, thus, likely to regulate TRAIL-induced cell death efficacy and modalities [109] and ultimately to affect immune antitumoral responses [110,111]. 

#### 2.1.4. TRAIL-Independent Induction of Apoptosis by DR4 and DR5 during Unfolded Protein Response

TRAIL agonist receptors have recently been found to contribute to unfolded protein response (UPR)-induced apoptosis, through the recruitment of the adaptor protein FADD and the caspase-8, in a TRAIL-independent manner [112,113,114,115,116,117]. The triggering of apoptosis during UPR is thought to be due, at least in part, to the binding of misfolded proteins to DR5, allowing the recruitment of the canonical pro-apoptotic machinery [118]. The contribution of DR4 and DR5 to UPR-induced cell death does not, however, appear to apply to all cell types, given that it was found to be negligeable in B-cell malignant cells [119].

### 2.2. Comparison of the Proximal Regulatory Mechanisms Governing TRAIL-Induced Cell Death with Other TNFRSF Members

#### 2.2.1. Intracellular Signalling Complexes Formation Induced by TNFRSF Members

Unlike TNF-R1 [120], engagement of apoptosis induced by DR4, DR5 or Fas is primarily initiated directly from the plasma membrane, through the formation of a complex coined Death-inducing signalling complex (DISC) [61,62,64] after TRAIL or Fas ligand binding to their respective cognate agonist receptors (Figure 2). TNFR1 membrane-bound complex, on the other hands, triggers a NF-kB-dependant survival pathway on the first instance, without recruiting FADD nor the caspase-8, due essentially to the recruitment of the kinase RIPK1 [121,122,123] and the adaptor protein TRADD [124,125] (Figure 2). 

The group of David Goeddel in the late 1990′s provided the first molecular demonstration that divergent signalling complexes could lead to distinct and antagonist signalling pathways [124]. While FADD and caspase-8 have long been known to be required for TNF-induced apoptosis [126,127], the molecular comprehension of their temporal and spatial contribution was unveiled, almost a decade later, by the discovery that a secondary complex was required to initiate apoptosis. Complex II is a soluble scaffold multimeric protein complex which arises from complex I [120]. It contains, amongst others, the adaptor protein FADD, the cysteine protease caspase-8, as well as the post-translationally modified forms of RIPK1 and TRADD, whose modification is primarily initiated in complex I [120]. Transition from complex I to complex II, albeit still not fully understood, was later on found to involve two proteolytic steps, starting first with the shedding of TNFR1 extracellular domain by TACE (TNF-alpha converting enzyme), also known as ADAM17 [128], and leading to the internalization of complex I through a clathrin-dependent mechanism, followed by an additional cleavage within TNFR1 transmembrane domain, by the γ-secretase, allowing the release of its intracellular domain, which contains bound TRADD, TRAF2 and RIPK1 amongst others proteins [128]. The release of complex I to the cytosol, in turn, subsequently allows the recruitment of FADD and caspase-8, forming the pro-apoptotic TNFR1-complex II (Figure 2).

#### 2.2.2. Regulation of TNFRSF Signal Transduction at the Extracellular Level

Regardless of the modus operandi required for engaging cell death by these receptors, the latter have been found to form dimers or trimers, due to spontaneous self-association of their N-terminal extracellular domain, called pre-ligand assembly domain (PLAD) [129,130], which is generally present in the first cysteine-rich domain of some TNFSFRs (Figure 1). By favouring ligand-independent receptor multimerizations, the PLAD limits apoptosis induced by TRAIL due to DR5 homodimerization [131], or the formation of heteromeric DR4, DR5, DcR1 or DcR2 complexes [42,56,132]. These self-association motifs have recently been demonstrated to be targetable. Interestingly it was found that, mere administration of a TNFR1 PLAD-Fc recombinant protein improves skin lesions in MRL/lpr [133], arthritis [134], as well as experimental autoimmune encephalomyelitis or diabetes [135], in experimental animal models. 

Organization and arrangement of TNFRSF in homo- and heteromeric complexes into higher-order complexes has profound effect on their signalling capabilities [15,136,137] and is often required for efficient apoptosis triggering, as demonstrated with DR5 [138,139,140]. Likewise, it has been proposed that, upon cognate ligand binding, DR4, DR5 and Fas form, first of all, trimer complexes whose multimerization or crosslinking with neighbouring trimers occurs via the dimerization between receptor interfaces, either located opposite the ligand-binding interfaces, resulting in a hexameric honeycomb-like structure [141]. A dimerization motif found in the transmembrane helix domain of the receptors is also suspected, in addition, to play an important role for the assembly of the DISC, its stability and potency [138,141,142]. Moreover, as suggested for Fas, DISC stability may also be regulated at the level of the cytoplasmic domain, of some agonist receptors, by the adaptor protein FADD [143,144,145].

Furthermore, in line with the fact that most TNFSF receptors harbour putative glycosylation sites, it has been demonstrated that O- and N-glycosylations, post-translational modifications, also regulate TNFRSFs pro-apoptotic signalling transduction [146,147]. Likewise, based on the observation that TRAIL sensitivity in cancer cells was associated with high glycosylation profiles, the seminal work of Avi Ashenazi’s laboratory, provided the first molecular demonstration that DR5-mediated TRAIL-induced cell death could be regulated by the O-glycosylation [148]. While it remains to be determined whether O-glycosylation affects other receptors of the family [149], receptors such Fas, TNFR1 or DR4 were found, on the other hand, to be N-glycosylated [150,151,152,153,154]. This post-translational modification of DR4 or Fas increases cancer cell lines sensitivity to TRAIL- or FasL-induced cell death, respectively [150,152]. Similar gain of function associated with the fly tumour necrosis factor (TNF) receptor homolog glycosylation were demonstrated [155]. It shall be noted, however, that N-glycosylation was found to prevent TRAIL-induced cell death in normal mouse fibroblastic cells [151], suggesting that the increase in signal transduction induced by TNFRSFR mediated by their O- or N-glycosylation, maybe restricted to cancer cells. Interestingly, the gain of function associated with the O- or N-glycosylation of these agonist receptors, with the exception of one study [155], is not related to a change in ligand binding to its cognate receptor, but rather to the stabilization of the membrane-bound primary complex, likely mediated by an increase in receptor aggregation, that ultimately leads to a better signalling activity, which in the case of Fas or TRAIL is associated with an increase in caspase-8 activation [148,150,152,156,157,158]. Consistent with this, glycan modifications or glycan-binding proteins were found to enhance or impair apoptosis induced both by TNFR1, FasL and TRAIL [149,157,159,160,161,162,163,164,165,166,167,168,169]. These post-translational modifications shall be distinguished from the recent findings reporting regulatory functions associated with the O-GlcNAcylations or O-GlcNAc. Contrary to the O- or N-glycosylation, O-GlcNAc takes place within the cytosol, and shall thus affect the C-terminal cytosolic domains of TNFRSFs. Likewise, there have also been reports demonstrating that GlcNAcylation of both DR4 or DR5 C-termini, could be required for, or enhance, DISC formation and receptor clustering [156,170,171]. On the other hand, O-GlcNAc of death-domain containing proteins, has also been demonstrated to protect cells, infected by pathogens, from apoptosis induced by TNFRSF-death-containing receptors [172,173,174], and erythrocytes from necroptosis by targeting RIPK1 [175]. Last, palmitoylation of DR4, Fas and TNFR1, another intracellular post-translational modification, was reported to enhance apoptosis induced by TRAIL [176] and Fas ligand [177,178,179] and to be required for TNFR1 signal transduction [180]. 

## 3. Physiological and Physiopathological Functions of TRAIL

TRAIL is expressed as cell surface protein, mostly by activated immune cells such as T and B cells [181], neutrophils [182,183,184], dendritic cells [185], monocytes and macrophages [186,187,188,189,190], natural killer and NKT cells (NK) [191,192,193,194,195,196,197,198,199,200,201]. TRAIL play a crucial role both during viral clearance [202,203,204,205,206,207,208,209,210,211,212,213,214] and tumour immune surveillance [215,216,217,218,219,220]. Mechanistically, during innate immunity, NK cells and CTLs (cytotoxic T cells) promote apoptosis of target cells, either by releasing soluble factors such as the cytolytic granules [184,221,222,223], which contain perforin and granzymes, or by engaging membrane-bound death ligands like FasL or TRAIL [200,221,224,225,226,227,228,229]. 

TRAIL exhibits pleiotropic physiological functions, which are regulated by its cognate receptors due to their ability to trigger or not cell death. TRAIL and its receptors play an important role in maintaining tissue homeostasis [230,231,232,233,234]. Through transducing cell death, TRAIL and its agonist receptors are most notoriously known for their ability to kill cancerous cells and cells infected by viruses [207]. Unlike FasL or TNFα [235,236], TRAIL induces apoptosis in tumour cells, selectively [237] and exhibits little to no cytotoxicity against normal human cells or murine cells [238,239,240,241,242,243,244]. Originally, TNFα was the first ligand of the TNFSF superfamily tested for its anti-tumoral activity [245,246], followed by Fas-ligand [236,247]. While Fas-ligand [52,248,249] and to a much lesser extend TNFα, due to the requirement of protein synthesis or transcription inhibitors [250,251,252], are efficient in killing a variety of tumour cells, these ligands cause significant damage to normal tissues that result in life-threatening toxicities [237]. Despite the fact that TRAIL, TNFα and Fas share common pro-apoptotic partners and modalities, only TRAIL displays tumour selective pro-apoptotic activity, sparing normal tissues or cells [237,244], including when administered to small animals or humans [243]. Likewise, administration of Fas or TNFα in rodents is lethal [236,253,254,255]. Moreover, TNF is involved in sepsis-mediated organ failure due to cellular toxicity [245,256,257]. Therefore, given that DR4 and DR5 are usually upregulated in cancer cells [258,259,260,261,262,263,264,265,266], and that TRAIL induces apoptosis in a p53-independent manner [267,268], contrary to most chemotherapeutic drugs [269], overcoming p53 escape [40], TRAIL attracted major attention in oncology [270,271,272,273]. 

Yet, a tremendous amount of work also suggests that TRAIL and its receptors are likely to play a role in several human diseases including, but not limited to, obesity and diabetes [274], and are associated with inflammation [2,117,275], neurological disorders [276] and cardiac diseases [277]. 

### 3.1. In the Immune System

In the immune system, TRAIL helps maintain lymphocyte homeostasis. Likewise, while activated CD8+ cells were described to be more sensitive than CD4+ T cells to TRAIL-induced cell death [278], CD8+ T cells can protect themselves from apoptosis induced by TRAIL by upregulating both the antagonist receptors and c-FLIP [193,279]. Variation in TRAIL sensitivity, in CD8+ T-cell blast, is both time- and stimuli-dependent, explaining TRAIL’s ability to actively contribute to CD8+ T-cell AICD and to generate memory-like CD8+ T cells [280,281,282,283,284,285,286]. Interestingly, using experimental animal models, TRAIL was found to inhibit autoimmune lymphoproliferative syndrome as well as spontaneous idiopathic thrombocytopenia purpura, due to its active contribution during activation-induced cell death (AICD) [285,287]. 

Besides its role in adaptative immunity, TRAIL plays an important role during in innate immunity [288], such as in anti-tumour immune surveillance [196,215,217,288,289]. TRAIL is often instrumental for the cytotoxic activity of immune cells. It is upregulated and contributes to the cytolytic activity of T cells, neutrophiles or monocytes stimulated by type I interferons [187,188,279], or after stimulation with IL-2 plus phytohemagglutinin [181], contributing to their anti-tumoral activity. TRAIL expression can also be induced in plasmacytoid dendritic cells by microbial or viral products such as LPS or Toll receptor agonists, contributing to their cytotoxic activity [290]. TRAIL is also thought to contribute to ocular [291] and placental immune privilege [292]. 

A recent study analysing the immune repertoire, in TRAIL-deficient mice, found organ-distribution differences in several types of immune cells, such as dendritic cells, in these animals as compared to parental mice [293]. Keeping in mind that CD8+ T cells were recently found to contribute to tissue remodelling [294] and that TRAIL can be expressed by a large number of immune cells, as mentioned above, including CD8+ cells, these studies collectively suggest that TRAIL may play a wider role in the immune system than expected. Indeed, growing evidence suggests that TRAIL non-apoptotic functions may also play a role in shaping and orchestrating the immune response to pathogens or cancer cells. TRAIL has, for example, recently been demonstrated to inhibit IL-15-induced cytotoxic granule granzyme B production in NK cells during viral infection, limiting viral clearance [207]. By regulating inflammation, in the absence of apoptosis, TRAIL contributes to the dysregulation of the immune system. Likewise, using TRAIL-R-deficient mice, it was found that TRAIL, by inhibiting T-cell activation, supress gut inflammation [295] or arthritis [296,297], in an apoptosis-independent manner [298]. Injection of TRAIL was also found to be beneficial in experimental animal models in inhibiting autoimmune thyroiditis [299] or arthritis [300]. Suppression of auto-immunity by TRAIL, can proceed both through a caspase-dependent and independent manner, as it was shown that TRAIL can, on the one hand inhibit Th1 cells proliferation, and on the other promote that of regulatory T cells, as demonstrated in TRAIL- [301] and TRAIL-R- deficient mice [296]. TRAIL-deficient mice also unveiled the critical role of TRAIL in supressing experimental autoimmune encephalomyelitis [302]. TRAIL functions in a remarkable way in autoimmune diseases by transducing both canonical and non-canonical signalling pathways, holding promises in autoimmune therapy [303,304]. Yet, in other instances, TRAIL has been found to trigger inflammation and/or to amplify other autoimmune diseases such as lupus erythematosus [305] and lupus nephritis [306]. 

Finally, TRAIL may also play a role in allergy, given that eosinophils and granulocytes express TRAIL receptors, but are insensitive to TRAIL-induced cell death [307,308], TRAIL is abundantly expressed in the airway epithelium, in response to allergen provocation, in the initial step [308,309,310].

### 3.2. In Diseases

TRAIL is associated with diseases beyond the immune system. Likewise, TRAIL may play a physiological role in endothelial cell function [311], since it has been found to exhibit a pro-angiogenic activity [312,313] and to stimulate the proliferation of vascular smooth muscle cells [314]. In another study, TRAIL, on the contrary, was shown to inhibit angiogenesis mediated by VEGF, through both a caspase-8-dependent and -independent manner [315]. In vivo, however, it was found, using *Trail−/−* mice, that TRAIL is able to promote angiogenesis and neovascularization after ischemia [316]. In the same line, an increasing number of studies also indicate that TRAIL could be involved during cell differentiation. Likewise, TRAIL induces the differentiation of intestinal cells [1], osteoblasts [317,318], skeletal muscle or myoblast cells [4,319] or keratinocytes [320], but appear to inhibit adipocyte differentiation [321]. 

TRAIL has also been described in lung and heart diseases. TRAIL induces the survival, proliferation and migration of human vascular smooth muscle cells (VSMC) in pulmonary arterial hypertension (PAH) [322,323,324]. Its high expression levels in the serum of PAH patients correlates with the severity of the disease [324]. Via non-canonical signalling, TRAIL promotes VSMC’s and fibroblast’s proliferation and migration through ERK1/2 MAPK and the serine/threonine kinase Akt activation, without affecting p38 MAPK or c-Jun N-terminal kinases (JNK) activation [325]. TRAIL stimulates proliferation of VSMC after insulin-like growth factor-1 receptor (IGR1) regulation via NF-κB activation [314]. In addition to VSMC, TRAIL promotes the survival and proliferation of primary human vascular endothelial cells, as well after Akt and ERK activation without affecting the NF-κB pathway [326]. Activation of NF-κB in vascular smooth muscle cells by TRAIL has also been described as requiring the cleavage of protein kinase C-delta (PKC-*δ*) by caspases [327]. 

TRAIL and its three receptors, DR4, DR5 and DcR1, are highly expressed in the human heart [261]. However, while cardiomyocytes express DR5, they are resistant to apoptosis, yet TRAIL was found to induce via DR5 the activation of the ERK1/2 pathway, in these cells, in a MMP-EGFR-dependent manner [328]. It has been proposed, in this study, that TRAIL, via inducing the production of MMPs, trigger the cleavage of the epithelial growth factor receptor ligand (HB-EGF) in the cell membrane to induce EGFR signalling, promoting cardiomyocyte proliferation and ERK 1/2 signalling [328]. Alternatively, although much less represented in the literature, other studies suggest that non-conventional ligand-to-receptor interactions may also exist, explaining how these agonist receptors may transduce non-apoptotic signalling pathways, such as the recently described soluble FasL/DR5 interaction, whose role during auto-antibody-induced arthritis has been associated with exacerbated inflammation in vivo through regulation of NF-κB-mediated production of CX3XL1 [329]. 

TRAIL’s pro-apoptotic or non-apoptotic signalling is also suspected to contribute at some extent to Alzheimer’s disease [330,331,332,333,334] and non-alcoholic fatty liver disease [335,336,337,338]. 

In most cases, the molecular mechanisms driving TRAIL-induced non-apoptotic signalling, including cell motility in normal cells or tumour cells remain poorly understood [24,117,339].

## 4. Signalling Machinery Associated with TRAIL Non-Canonical Transduction

TRAIL, as reported in a growing number of studies, trigger the differentiation, proliferation and survival of normal cells, such as macrophages [2,317], intestinal mucosal cells [1], skeletal myoblasts [319], keratinocytes, osteoclasts [318,340], vascular smooth muscle cells [4,325,326,341] and mouse fibroblasts [342]. 

In cancer cells, on the other hand, if apoptosis is not efficiently triggered, TRAIL can be detrimental to patients given that this cytokine also exhibits pro-tumoral properties, associated with TRAIL’s ability to induce inflammation, tumour cell motility and invasion, ultimately leading to metastasis [8,11,12,114,343,344,345,346,347]. Likewise, TRAIL was found to promote the proliferation in human glioma cells through ERK1/2 phosphorylation and the stabilization of the long form of c-FLIP(L) [348], in cholangiocarcinoma cells via NF-kB [13]. Migration and invasion were also promoted by TRAIL in NSCLC the A549 cell line in a RIPK1-dependent manner through phosphorylation of Src and STAT3 [12], in pancreatic ductal adenocarcinoma [11], in colorectal cancer cells, resistant [346] or not [114] to TRAIL-induced cell death and in the triple-negative breast cancer cell line MDA-MB-231 (TNBCs) [114]. In oesophageal squamous cell carcinomas (Figure 5), TRAIL induced epithelial–mesenchymal transition (EMT) and metastasis through ERK1/2 and stat3-dependent upregulation of PD-L1 [349]. PD-L1 regulation via ERK phosphorylation induced by TRAIL was also reported in TNBCs [289]. Using a TNBC xenograft model, TRAIL was also demonstrated to promote skeletal metastasis [347]. Consistent with these findings, deletion of murine TRAIL-R, in a non-small-cell lung cancer (NSCLC) and pancreatic ductal adenocarcinoma (PDAC) using a KRAS-driven experimental model, was found to drastically impair metastasis, and this effect was associated with a loss of cell migration, proliferation and invasion [8].

Mechanistically, TRAIL was shown to induce NF-κB activation [20,350,351] and by analogy with TNFR1 signalling [120], albeit in a distinct manner, it was then found that TRAIL could lead to the formation of two main distinct molecular complexes, explaining, at least in part, how TRAIL receptors can transduce cell death or pro-inflammatory pathways [12,345]. The primary pro-apoptotic complex, known as TRAIL DISC, is mostly composed of the TRAIL receptors, FADD, caspase-8 or -10 and the inhibitor c-FLIP, and is localized at the level of cellular membranes [42,61,64,352,353]. RIPK1 is also present in this complex [24,354] as well as TRADD [20,350,355,356]; however, there might be some differences in TRADD binding to TRAIL receptors, given that TRADD seems to be preferentially recruited to DR4 [20,350,354]. In addition to these adaptor proteins and kinases, originally found to compose TRAIL membrane-bound complex, kinases such as IKKα, IKKβ and IKKγ are recruited to complex I, explaining how NF-κB may be induced by TRAIL [357]. Native recruitment of ubiquitin ligases can also happen in the TRAIL DISC as demonstrated with the presence of the linear ubiquitin chain assembly complex LUBAC (Figure 2), whose components SHARPIN and HOIP limits TRAIL-induced cell death as well as NF-κB activation [357,358,359], due to RIPK1 and FADD linear ubiquitination [357]. Moreover, other proteins such as c-IAPs, A20 and TRAF-2 are also recruited in complex I [357].

The secondary non-apoptotic complex, on the other hand, is found in the cytosol, while it arises from complex I [358] (Figure 3). Complex II contains not only FADD and caspase-8, but also RIPK1, TNF receptor-associated factor 2 (TRAF2), TRADD, as well as a large number of apoptosis inhibitors, NF-κB regulators, including IKK and NEMO [345], not to mention LUBAC [357] (Figure 4). It must be stressed here that RIPK1 can not only be directly recruited to TRAIL receptors, as evidenced in native complex I [357,360,361], because it contains a death domain [362], but that the latter is required for TRAIL-induced NF-κB activation [363]. Of interest, similarly to Fas DISC [94], membrane-proximal localization of RIPK1 allows its cleavage by the initiator caspase-8 within its intermediary domain, abolishing TRAIL-induced NF-kΒ activation [360,361].

Given that RIPK1 is recruited to the TRAIL DISC and present in the cytosolic complex II, it is easy to understand how TRAIL triggers the NF-κB pathway. Yet, as was demonstrated more than 10 years ago by Azijli and co-workers, in the TRAIL-resistant cancer cell line A549, TRAIL induces, in addition to NF-κB, the phosphorylation of a large number of substrates associated with activation of the P38, ERK1/2, JNK1, Src, AKT, Raf1 and ROCK [364]. While the implication of TRADD for TRAIL signalling is less investigated, TRADD was found to afford protection against TRAIL-induced apoptosis [355,365,366], but more interestingly TRADD was suggested to play an important role in the secondary complex to induces IL-8 secretion in NSCLC, under TRAIL treatment [367]. Furthermore, TRADD and RIPK1 redundantly mediate proinflammatory signalling in response to TRAIL in human ovarian HeLa metastatic cell line [354]. Despite the fact that several experimental evidence link for example ERK1/2 activation in glioma cells with c-FLIP [348] or JNK activation with RIPK1 [363], it remains unclear how upstream kinases are integrated and activated in the molecular platforms triggered by TRAIL, whether it be complex I or complex II. 

Evidence has accumulated demonstrating that TRAIL can be detrimental in oncology, due to its ability to promote cell migration and metastasis; it still remains unknown, however, whether both TRAIL-agonist receptors trigger similar non-canonical signalling activity. Unlike rodents [38], primates express two TRAIL agonist receptors [31,34,36]; therefore, it should be kept in mind that the findings obtained from genetically modified mice may not transpose to primates. For instance, with the exception of one study [368], migration and metastasis promoting TRAIL’s activity seem to be mostly associated with DR5 [8,12,114,347]. While it remains unclear whether this peculiarity is due to DR5 splice variants or not [369], DR5 is found to be overexpressed in several cancer types and this overexpression is often associated with tumour aggressiveness and poor patient prognosis [370]. For example, DR5-positive staining is associated with increased risk of patient death in non-small cell lung cancer [260], breast [264] and renal cancer [371]. 

Activation of this non-canonical signalling pathway by DR5, which promotes tumour growth and metastasis through MAPK, PI3K/AKT or NF-κΒ signalling, is likely to be only visible in TRAIL-resistant cancer cells [12,346], including cells expressing TRAIL decoy receptors [59,372]. Alternatively, transition of the receptors once engaged with the ligand to membrane lipid rafts, may as demonstrated for TNF [373], contribute to induction of the pro-migratory signal. It has been suggested for example, that lipid rafts may provide an adequate membrane platform for aggregation for DR4/DR5 to transduce apoptosis [374]. Localization to lipid raft may be differentially occurring depending on the receptor and its potential palmitoylation status. Likewise, DR4 can be palmitoylated, translocating to lipid raft, where it was proposed to form and activate the pro-apoptotic complex I [176]. In B-cell hematologic malignant cells, DR4 was even proposed to be constitutively localized within lipid rafts [375]. While DR5 was not found to be palmitoylated, it has been described in lipid raft and proposed to recruit and activate the caspase-8 in these subcellular compartments [374,376,377,378,379]. However, while there is no doubt that TRAIL complex I may transit to lipid rafts, native TRAIL DISC formation in these lipid-rich structures have never been demonstrated. On the contrary, it was found that TRAIL DISC-mediated activation of the initiator caspase-8, which is required for initiating apoptosis, instead occurs in non-lipid rich membranes [42,380]. Nonetheless, it cannot be excluded that transient translocation to lipid raft may account for TRAIL’s pro-tumoral properties.

### 4.1. Lessons from Fas/CD95-Induced Non-Canonical Signalling (Secondary Complex)

Non-canonical pro-motile and pro-metastatic signalling was also documented for Fas, a receptor of the TNF superfamily, which, like DR4 and DR5, is able to engage apoptosis from the membrane in a FADD- and caspase-8 dependent manner [381]. Fas ligand (FasL) was found to redistribute its agonist receptor, Fas, dynamically into lipid rafts, contributing to the elimination of activated T cells [382]. Lipid rafts were, thus, soon considered as possible check point controls for FasL-induced Fas signalling cellular outcome [383,384]. Like TRAIL, but to a much lesser extent than TNFα, FasL is also able to transduce NF-κB, regardless of its ability to trigger apoptosis [25,385]. NF-κΒ activation by FasL was associated with resistance to apoptosis in cancer cells [16], but also appeared to be associated, in addition, to cell motility and invasiveness [29]. It was also demonstrated that naturally cleaved FasL could induce cell migration [386,387,388]. Fas was found to induce proinflammatory cytokines in human monocytes [26,389]. In dendritic cells, Fas stimulation induce IL1β and IL-12 production and cell maturation [390]. 

Mechanistically, it remains unclear how Fas induces cytokine production or how it activates its pro-metastatic signalling pathway. FasL-induced cell motility and invasion has been associated with TRAF2 [391], PDGFR-β-mediated PLC-γ1 activation and PIP2 hydrolysis [392], activation of the kinase c-Yes and AKT and changes in cytosolic calcium [386], Rac1 [393] or via phosphorylation of Rock1 and involvement of the Na^+^/H^+^ exchanger NHE1 [388]. TRAF2 is recruited within the TRAIL DISC [357,394]. By allowing recruitment of ubiquitin ligases within the primary complex, TRAF2 is able to limit caspase-8 activation [357,394,395]. TRAIL-induced JNK activation was found in cancer cell lines to require RIP and TRAF2 [396], suggesting that many of the non-canonical signalling pathways may be readily engaged from complex I. Alternatively, it has recently been proposed that NF-κΒ-mediated initiation of inflammation upon TRAIL stimulation may be induced, at least in part, through TRAF-2-mediated recruitment of cIAP1/2 and LUBAC into complex I, leading to the formation of a secondary complex coined ‘‘FADDosome’’ in which RIPK1 undergoes linear ubiquitination, allowing assembly of the NF-kB machinery and NF-κΒ-dependent regulation of inflammatory cytokines and chemokines [397] (Figure 5). 

Linear ubiquitination and stabilization of the NF-κB signalling by LUBAC was first uncovered in TNFR1 complex I and found to rely on TRADD, whose absence precludes both TRAF2 and LUBAC recruitment to TNFR1 [398], consistent with the need of TRADD to induce NF-κB activation by TNFR1 [125] and to allow TRAF2 recruitment to TNFR1 [124]. Within the Fas DISC, the caspase-8 inhibitor c-FLIP was also found in the early days as a protein that could integrate at TRAF2, to induce both NF-κB and ERK signalling [399,400]. Keeping in mind that TRADD could be essential too, for TRAIL-mediated non-apoptotic signalling, including induction of NF-κB [354,366], it is worth mentioning that TRADD is found both associated with TRAIL receptors membrane complex I [350,357] and soluble complex II [345]. An alternative molecular circuitry may explain the biological activity of TRAF2 in driving TRAIL pro-tumoral effects. Likewise, it was described that NF-kB activation by TNFR1 requires sphingosine-1-phosphate (S1P). S1P interacted with TRAF2 as a co-factor to catalyse RIPK1 poly-ubiquitination and NF-κB activation [401]. Given that S1P may be critically linked to metastasis [402,403], it may be worth considering, in addition, the interesting work demonstrating that deletion of DR5 induce cell motility and promotes cell invasion in a TRAF2 and S1P-dependent manner, through activation of the JNK/AP-1 pathway in lung cancer cells [368,404] (Figure 4). 

Direct recruitment of kinases associated with non-apoptotic Fas signal transduction as also been found, including Rac1 activation after binding to Fas membrane proximal domain (MPD), located in the intracellular part of the receptor, during neurite growth [393]. While not characterized molecularly, TRAIL-induced cell motility was also associated with Rac1 activation in monocytes [405] and HeLa cells [406]. Interestingly, though, while Rac1 appears dispensable for the regulation of inflammatory proteins after TRAIL stimulation [407], Rac1 was required for DR5-mediated cancer cell motility and metastasis [8], and similar to Fas, the MPD of DR5 was also suggested to be required to trigger this effect (Figure 5). Consistent with mutated KRAS’s ability to inhibit ROCK1 [408], ROCK1 inhibitors allowed Rac1 recruitment to DR5 and transduction of a signalling pathway leading to invasion in non-mutated KRAS cells [8]. It is thus likely that direct recruitment of RAC1 into the TRAIL DISC, due to its ability to promote filopodia and lamellipodia formation, may lead to microtubules and cytoskeleton rearrangements [409], accounting for the cell migration induced by DR5 [8] (Figure 5).

### 4.2. Calcium Signalling Inducing Cell Motility and Metastasis

Calcium signalling induced by ligands of the TNF family has initially been addressed with TNF [410] and FasL [411]. Increased cytosolic Ca^2+^ was found to occur almost immediately after stimulation, within the first 50 s. High calcium levels have been recorded after stimulation by FasL following activation of phospholipase Cγ1 (PLCγ1), inositol 1,4,5-trisphosphate (IP3) generation, IP3 receptor (IP3R) calcium ionic channels stimulation and a late secondary Cytochrome-c-triggered activation of endoplasmic reticulum (ER)-resident calcium channels [412]. The role of Ca^2+^ in cancer cell proliferation, migration and invasion has been well established [413]. Likewise, Ca^2+^ signalling is a potential key regulator for breast cancer bone metastasis and prostate cancer cells proliferation, angiogenesis, EMT, migration and bone colonization [414]. Interestingly, both TRAIL- and FasL-induced pro-metastatic pathways are associated with an early increase in intracellular Ca^2+^ and tyrosine kinase signalling [114,415,416]. The use of isogenic stable cancer cells deficient for either DR4 or DR5 [114], demonstrated that TRAIL-induced pro-metastatic signalling was solely triggered by DR5 and correlated with a rapid Ca^2+^ flux [114,417,418]. Furthermore, early increased cytosolic Ca^2+^ was shown to be activated upon TRAIL exposure in both Jurkat and NB4 leukemia cells, protecting the latter from apoptosis [418]. It was found in these cells that recruitment of both p62 and ATG7 to complex I was required for calcium influx induced by TRAIL [418]. 

Like TRAIL, FasL also induces an increase in cytosolic Ca^2+^, associated with cell-motility and metastasis [386,415,419,420,421]. Intracellular increase in Ca^2+^ is generally induced by PLCγ1 and IP3R activation, due to ER Ca^2+^ release [422], but may also be triggered, as demonstrated in leukemia cells, after ORAI1 activation and CRAC channels opening [418]. Autophagy Related 7 (ATG7) [423] and Sequestosome 1 (p62/SQSTM1) [424], are two autophagic proteins related to ORAI1 and CRAC channels, whose recruitment to DR5 induce the release of Ca^2+^ from the ER [418]. Keeping in mind that DR5 is also involved during apoptosis induced during the ER stress and that this process is associated with Ca^2+^ release [112,118], while DR5, but not DR4, is able to induce a change in calcium flux after TRAIL stimulation, these findings suggest that calcium regulation is probably important for the triggering of TRAIL-mediated non-apoptotic signalling. Indeed, FasL also induces high intracellular levels of Ca^2+^ ions to promote, depending on the context and cancer cell type, apoptosis or non-canonical signalling [412]. How Fas or DR5 trigger these changes in intracellular calcium remain unknown. However, in two studies performed using breast cancer models, DR5 was proposed to directly interact with a protein which has a calcium dependent activity, the calmodulin (CaM) [425,426] (Figure 5). CaM is a small Ca^2+^ binding protein that interacts with a large group of intracellular proteins and which participates in signalling pathways that regulate proliferation and motility [427,428]. In PDAC cells, CaM was also found to be recruited in the DR5 DISC together with c-FLIP and the proto-oncogene Src, contributing to cell resistance [429]. In NSCLC cells, CaM inhibition or Ca^2+^ deprivation inhibited the recruitment of Src and was associated with an increase in c-FLIP short degradation, sensitizing cells to DR5 agonist-induced apoptosis [430]. Src could play a role during TRAIL-induces non-canonical signalling [12], given that Src was described, in addition, to phosphorylate and, thus, to inhibit caspase-8 enzymatic activity [431]. Furthermore, CaM may allow recruitment and activation of the Src [432]. Interestingly, CaM has also been found to be recruited within the Fas DISC [421,433,434], and associated with the regulation of a Src pro-tumoral activity [421,432]. Last, caspase-8, alone, was found to bind to the focal adhesion kinase (FAK) and calpain-2 Ca^2+^ dependent protease (CPN2), displaying pro-metastatic function properties in glioblastoma cell lines [435], (Figure 5). 

### 4.3. Nuclear DR5 Regulates Both Proliferation and Metastasis

In other studies, regulation of TRAIL’s pro-tumoral signalling has been suggested to be due to the subcellular compartmentalization of DR5 in the nucleus [436,437]. It is not clear how DR5 goes to the nucleus, but it has been proposed that DR5 may undergo proteolytic cleavage or internalization upon ligand binding, allowing its translocation into the nucleus [438,439,440]. Interestingly, mostly DR5 but not DR4 is found in nuclear compartment in late cancer stage of NSCLC [437], pancreatic [441] and breast cancer [442]. DR5 harbours two nuclear localization signals (NLS) sequences which promote importin-β1 binding and nuclear translocation of the complex, thus limiting TRAIL-induced cell death sensitivity [439]. In the nucleus, importin-β1/DR5 was found to regulate the micro-RNA let-7 maturation and to promote tumour cell proliferation [441]. 

Mature let-7 is known to control cell proliferation by inhibiting its targets, such as, the High mobility group AT-Hook protein-2 (HMGA2) and the Lin-28 homolog-B (Lin28B) protein expression. Upregulation of HMGA2 and Lin28B enhance cell proliferation and malignant progression [443,444,445,446] (Figure 4). HMGA2 and Lin28B are two proteins overexpressed in embryonic tissues and downregulated in differentiated tissues because of low expression of let-7. Let-7 overexpression prevents cell transformation in epithelial cells [447]. Furthermore, silencing DR5 using shRNA leads to increased levels of mature let-7, which in turn results in lower levels of let-7 targets, and reduced cell proliferation in pancreatic cancer cells [441]. Interestingly, knockdown of DR5 in metastatic breast cancer cells decreases bone homing and early colonization to the bone marrow and induces E-cadherin overexpression in xenograft mice model [347]. Impaired cell migration was linked to decreased CXCR4 expression [347] and increased E-cadherin expression [448]. CXCR4 selectively binds the CXC chemokine stromal cell-derived factor-1 (SDF-1), also known as CXCL12, and plays a crucial role in several biological processes, including in cancer biology, where it was associated with tumour dissemination and metastasis [449]. CXCR4 is a marker of breast cancer cells poor prognosis. High CXCR4 expression is significantly correlated with lymph node status, distant metastasis and poor survival [450]. Interestingly, nuclear DR5 regulates CXCR4 expression through inhibiting let-7 maturation [14,347], leading, as a consequence, to the expression of HMGA2 and CXCR4, and bone metastases formation of breast primary tumours [347,441,451] (Figure 4). All these findings suggest that nuclear DR5 may also play an important function in tumour aggressiveness. Yet, whether translocation of DR5 to the nucleus is fast enough to explain and concur to calcium-mediated pro-motile and metastatic signalling after TRAIL treatment, remains to be determined?

### 4.4. Caspase-8 Contribution in TRAIL Non-Canonical Signalling

Caspase-8 and FADD are required for TRAIL to be able to induce apoptosis and are both recruited to TRAIL DISC upon TRAIL treatment [61,62], but recent evidence suggest that they may also contribute to TRAIL non-canonical signalling. Likewise, caspase-8 has been reported to be recruited to a FADDosome complex, whose formation after TRAIL stimulation is associated with cell proliferation and/or migration [397]. Interestingly, mutations of caspase-8 in head and neck squamous cell carcinomas represent almost 9% of the cases, and three out of the four mutations examined in Li’s study conferred caspase-8 with pro-motile and pro-invasive properties [452]. Moreover, phosphorylation of caspase-8 on tyrosine 380 by the Src kinase, which inhibits its aspartate protease activity and, thus, protects cells from TRAIL-induced cell death [431], was associated the likelihood of a regulation of caspase-8 functions, switching its pro-apoptotic activity to cell migration by SH2 kinases [453,454]. Caspase-8 Y380 residue was described to be essential for caspase-8 relocalization to lamella of migrating cells [455]. Src-induced phosphorylation of caspase-8 on Y380 was also found to drive the assembly of a soluble complex, containing IKKα, IKKβ and p65, that tiggers NF-kB activation in glioblastoma cells, leading to inflammation and angiogenesis [456]. 

Caspase-8 has been described to interact with p85α, subunit of PI3K to activate Rac1 through lipid products generation (PIP2 and PIP3) that activate guanine nucleotides-exchange factors (GEFs), [457] which are necessary to Rac1 activation [458]. In neuroblastoma cell lines caspase-8 pro-migratory signalling capability was associated with its ability to interact with the focal adhesion kinase (FAK) and calpain 2 (CPN2) [459], two components of the focal adhesion complex (FAC) [435] (Figure 5). FAC is a signalling complex anchored by cell actin cytoskeleton, membrane integrins and extracellular matrix (ECM). This complex is known to contain many cytosolic proteases, phosphatases, and kinases, including the FAK, a key effector of metastasis [460]. Cytoplasmic phosphorylated FAK induce cell migration and invasion, cytoskeleton organization and EMT through FAC protein elements activation, like PI3K, Src and Rho [461]. Caspase-8 interacts with components of the FAC in a tyrosine-kinase dependent manner, promoting both cell migration and metastasis [453,461,462]. Of interest, it was also found that FADD, by inhibiting miR7a expression, is associated with an increase in FAK and spontaneous invasion and metastasis of the melanoma cell line B16 [463]. The increase in FAK overexpression, induced by a FADD-mediated downregulation of miR7a, leads to the expression of CCL5 and TGFβ, two cytokines involved in triggering metastasis [463,464] (Figure 5). Last, but not least, caspase-8 pro-motile and metastatic signalling has also been associated with its ability to promote Rab5-mediated internalization and recycling of β1 integrins [465,466].

Consistent with the findings described above and the work of Henry et al. [397], indicating that both FADD and caspase-8 may account for TRAIL non-apoptotic signalling, is the demonstration, in rheumatoid arthritis fibroblast-like synoviocytes, that caspase-8 is responsible for the cellular migration of these synoviocytes stimulated with PDGF, regardless of its enzymatic activity [467]. 

### 4.5. TRAIL Induce Cancer Metastasis after uPA and c-cbl Regulation

TRAIL was found to enhance inflammation and promote invasion of PDAC cells in vitro and metastasis in vivo by inducing the upregulation of the urokinase-type plasminogen activator (uPA), IL-8 and CCL2 [11]. uPA is an agonist of the urokinase-type plasminogen activator receptor (uPAR) which can induce metastasis [468]. It has been found to be involved in triggering FasL-induced invasiveness [29]. uPA converts plasminogen to plasmin, then activates MMPs under matrix extracellular degradation [469]. Activated uPAR can also, on the other hand, interact with other transmembrane receptors, including integrins and growth factor receptors [470,471,472]. These interactions trigger activation of the ERK1/2, FAK, Src and PI3K/Akt signalling pathways [473,474]. 

Besides regulating metastasis, uPAR was found to inhibit TRAIL-induced apoptosis by regulating, the expression of DR4 and DR5 in glioma cells [475], the intrinsic mitochondrial pathway in colon cancer [474] or miR-17 and miR-20 expression levels in TNBC, two miRNAs that were shown to impair DR4 expression [476]. Using a RAS-derived stepwise tumorigenesis model to recapitulate TRAIL selectivity, Pavet et al. demonstrated that PLAU mRNA levels, encoding uPA, increase during transformation, preventing TRAIL-induced apoptosis [477]. Depletion of uPA restored TRAIL sensitivity, through inhibiting ERK1/2 activation and DcR2 recruitment to the TRAIL DISC [477]. Mechanistically, how uPA/uPAR regulate TRAIL signalling and more specifically cell motility and metastasis is still unknow. Yet, given that uPA is known to promote, not only cancer cell survival or proliferation, but also migration from primary tissues to distant organs [478], it remains an interesting potential TRAIL receptor complex partner to study. 

In addition to uPA, the ubiquitin ligase Cbl proto-oncogene (c-Cbl) has also attracted attention as a potential TRAIL-receptor partner for the triggering of TRAIL pro-metastatic signalling. This ubiquitin ligase was found to regulate both DR5 and DR4 expression levels [479,480,481]. c-Cbl was found to interact with the caspase-8 inhibitor c-FLIP and to induce its proteasomal degradation, sensitizing macrophages, infected by mycobacteria, to TNF-induced cell death [482]. A number of studies point to c-Cbl as a potential regulator of TRAIL non-canonical signalling pathways [483,484,485]. Likewise, after TRAIL stimulation, c-CBL appears to be involved in a complex involving Src and PI3K, which induces the phosphorylation of AKT [483]. CBL-b and c-CBL were found to interact with DR5, linking DR5 with TRAF2 and inducing ubiquitination of caspase-8 in TRAIL resistant gastric cancer cells [394]. CIN85 is an important c-Cbl binding protein which plays an essential role in cell survival [486], such as for example in prostate adenocarcinoma cells, in which CIN85 was found to enhance the phosphorylation and activation of MAPKs during TRAIL treatment, leading to their survival [485].

Interestingly, while only cell death was analysed in Xu et al.’s study, it was also found in these cells that deletion of CBL-b restored TRAIL sensitivity and also had an impact on TRAIL-receptor subcellular localization [484,487]. Besides TRAIL agonist receptors, it was found that activated c-Cbl induce EGFR redistribution into lipid rafts, facilitating its activation [484], which might ultimately promote metastasis in gastric cancer cells (Figure 5). 

## 5. Conclusions and Perspectives

TRAIL has emerged as a promising anticancer agent; however, resistance to TRAIL is a major problem, not only because targeted tumours will likely survive to the treatment, but most of all because TRAIL may trigger, in resistant cells, a non-conventional signalling pathway that may ultimately lead to tumour spreading and metastasis. 

While signalling pathways triggering cell death are well understood, non-canonical signalling pathways driving cell motility and leading to metastasis are still unclear. As discussed in this review, a number of molecular complexes have been described, explaining how TRAIL receptors may drive cell survival, proliferation, inflammation and metastatic signal transduction. Yet, it is still unclear whether NF-κB or MAP Kinase signal transduction requires a secondary complex or not, given that main kinases or adaptor proteins, including RIPK1, TRADD or TRAF2 can readily interact with complex I. Comprehension of both the temporality and the subcellular localization and composition of these complexes is still missing to provide a comprehensive view of the molecular circuitry which dictate pro-apoptotic or non-apoptotic signalling pathways triggered by TRAIL receptors.

Regardless, a better understanding of the molecular events involved during TRAIL-induced pro-metastatic signalling or non-apoptotic signalling pathways would be beneficial for both cancer therapies and auto-immune diseases, as this will likely open interesting opportunities to prevent autoimmune diseases associated or not with inflammation or to inhibit or cure metastasis formation in patients.

## Figures and Tables

**Figure 1 cells-13-00521-f001:**
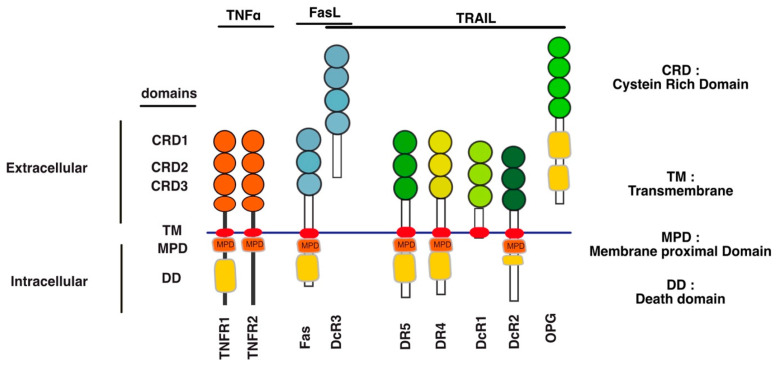
Schematic representation of TNFRSF sub-family receptors binding to TNFα, FasL and TRAIL. Receptors are depicted with their three main functional domains. The extracellular domain of these receptors is composed of cystein rich domains (CRD), orange for TNFR1/2; blue for Fas and DcR3 and a panel of greens for DR4, DR5, DcR1, DcR2 and OPG. Their TM (transmembrane domain) is represented in red, whereas their intracellular domains, with the exception of DcR3 and OPG, which are secreted, is represented by a bar (solid or not). Some of these receptors harbour in addition a death domain (DD), represented as a yellow box. Note that the DD of DcR2 is truncated. The solid bar above each ligand encompasses the receptors with which a physical interaction has been demonstrated experimentally.

**Figure 2 cells-13-00521-f002:**
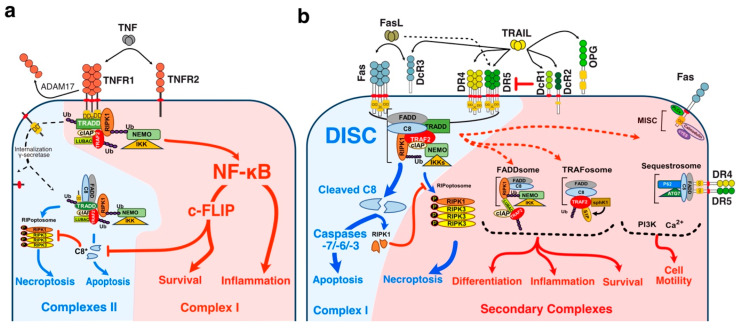
Comparison of the signalling pathways triggered by TNFR1, Fas and TRAIL agonist receptors. (**a**) TNFR1 signalling complexes upon TNFα stimulation are depicted in this panel (see also the text). TNFR1 engages first of all the formation of complex I, a survival membrane-bound platform which leads to the activation of the NF-κB pathway, and, in most cases, to cell survival and inflammation. Regardless of the outcome, complex I is processed during activation to give rise to a secondary soluble complex (complex II), that recruits pro-apoptotic components such as the adaptor protein FADD and the caspase-8 to induce apoptosis. Cell death is usually never happening, unless activation of the NF-kB pathway fails, because the latter induce the transcriptional regulation of cellular FLIP (c-FLIP), the main inhibitor of caspase-8. (**b**) Engagement of Fas, DR4 or DR5, contrary to TNFR1 enable direct recruitment of FADD and caspase-8 at the membrane in complex I. These agonist receptors are thus more prone in triggering apoptosis than TNFR1. Non-apoptotic signal transduction, however, is thought to proceed from a secondary complex coined complex II, which has been described as the FADDosome, Sequestrosome or the MISC (migration signalling complex). The latter leads to the activation of the NF-κB pathway to induce survival, pro-inflammatory and pro-tumoral effects. The secondary necroptotic complex is also depicted for TNFR1, TRAIL receptors and Fas (see text for more details).

**Figure 3 cells-13-00521-f003:**
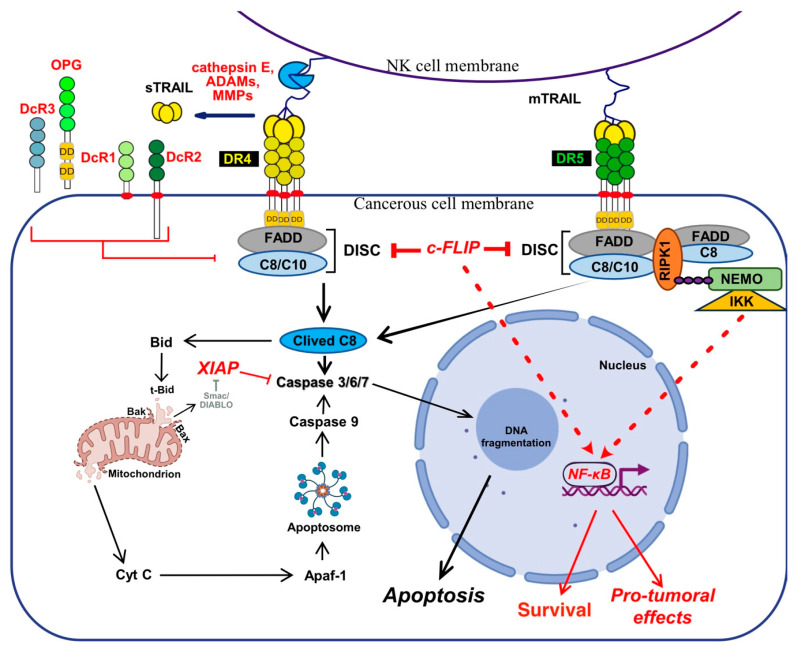
Schematic representation of TRAIL canonical signalling pro-apoptotic pathway. Illustration of the membrane-bound TRAIL, expressed by cytolytic immune cells such as NK cells inducing apoptosis in cancer cells. TRAIL binding to DR4 and/or DR5 agonist receptors, induce their aggregation and the recruitment of FADD and caspase-8/10 forming the DISC (death-inducing signalling complex), or complex I, which ultimately will lead to the activation of the effector caspases 3/6/7. Their activation by enzymatic cleavage is either triggered directly by the active caspase-8 or indirectly through caspase-8-mediated Bid cleavage. Bid cleavage allows Bax translocation to mitochondria and the release of cytochrome c, whose binding with Apaf-1, amplifies apoptosis-induced by TRAIL receptors (extrinsic pathway), through the formation of a soluble pro-apoptotic complex coined apoptosome, that allows activation of the initiator caspase-9. The active caspase-9 will in turn amplify the signal by cleaving and activating the effector caspases 3/6/7. The main inhibitors of this signalling pathway are represented in red, including the antagonist receptors (DcR1/2/3 and OPG) which compete for TRAIL binding or c-FLIP and XIAP the main caspase-8 and effector caspases inhibitor inhibitors, respectively. In addition, a schematic representation of the non-canonical signalling associated with complex I is shown, mainly describing potential activation of NF-κΒ pathway, which besides protecting the cells from TRAIL-induced apoptosis is involved in promoting TRAIL’s pro-tumoral activity. Main TRAIL-induced apoptosis inhibitors are shown in red.

**Figure 4 cells-13-00521-f004:**
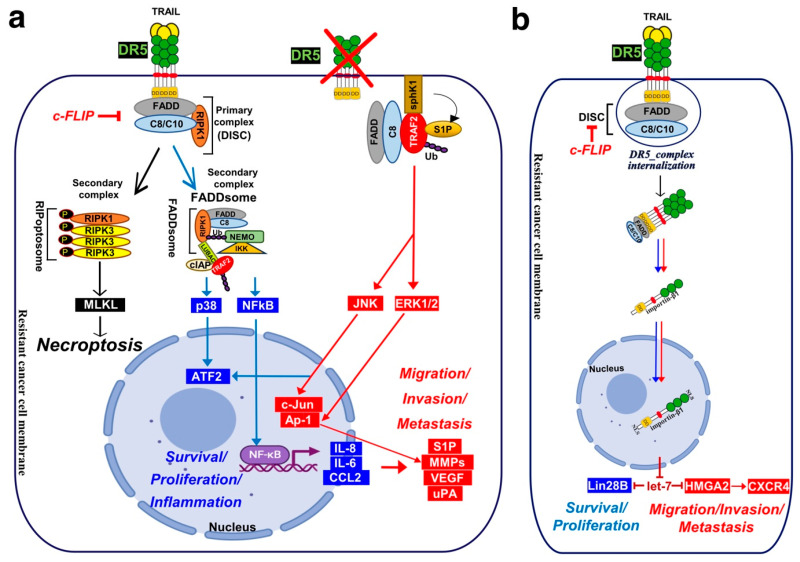
Schematic representation of DR5 non-apoptotic signalling pathways. Illustration of (**a**) DR5-mediated RIPoptosome and FADDosome secondary complexes and (**b**) nuclear translocation of DR5 in the nucleus, potentially mediating cell migration. See text for explanation.

**Figure 5 cells-13-00521-f005:**
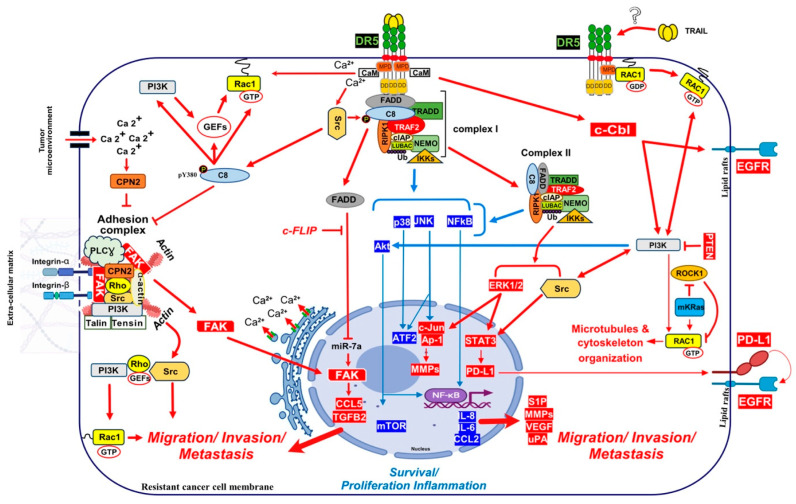
TRAIL-induced non-canonical pro-tumoral signalling via DR5 secondary complex formation. TRAIL agonist receptors, especially DR5, depending on cancer type and stage, can promote tumour growth and metastasis either through complex I or through a soluble secondary complex. Complex II arises from complex I and contains amongst other FADD, caspase-8, RIPK1, TRAF2, TRADD, cIAP, LUBAC, NEMO and IKKs. While complex I is associated with survival and proliferation through p38, JNK and NF-kB activation. Complex II appears in addition able to activate ERK1/2 pathway and Src leading to metastasis in vivo (see text for explanations). DR5 can directly activate signalling proteins involved in metastasis, thanks to its membrane-proximal domain (MPD), represented in orange, which directly recruits a Ca^2+^-binding protein, the CaM whose recruitment, in the presence of calcium, induce the activation of the proto-oncogene Src and the ubiquitin ligase c-Cbl, leading to PI3K, JUN, STAT3 and Rac1 activation. Activation of Rac1 promotes microtubules and cytoskeleton organization to activate cell migration. Activated CaM, by inducing Src activation can induce the phosphorylation of the caspase-8 on tyrosine 380, see text for details, enabling PI3K activation and subsequent activation of Rac1, leading to cell migration and invasion. Caspase-8 phosphorylation can also inhibit the adhesion complex through interacting with CPN2. This interaction inhibits cell adhesion and allows complex elements activation. Cell migration and invasion can next be induced by activating FAK and additional adhesion complex elements, such as PLCy, Rho, PI3K and Src. FADD, another DISC component has been described for its ability to trigger FAK by inhibiting miR7a expression via unknown mechanisms. This inhibition is linked to the expression of the pro-metastatic cytokines TGFβ and CCL5. Rac1 is also found, as illustrated here to be activated by direct recruitment to DR5 MPD, in a ligand-independent manner, but may also be activated indirectly (See text for additional details). Colours: writing highlights and arrows illustrate TRAIL-induced proliferation and inflammation (in blue), or TRAIL-induced metastasis (in red).

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
