# Peer review of "TNF-Related Apoptosis-Inducing Ligand: Non-Apoptotic Signalling"

_cells, 2024, doi:10.3390/cells13060521_

Round 1

Reviewer 1 Report

Comments and Suggestions for Authors

This is an excellent and comprehensive review of non-canonical TRAIL-receptor signalling. It has a number of informative figures and in my view the article will be of great interest to the scientific community. 

I just have a few minor comments:

Overall, there are a number of typos in the manuscript. A few examples are shown below.

1.       The unfolded-protein response and the involvement of TRAIL-receptor needs to be edited between lines 199 and 211. It is not very clear and hard to follow at the moment.

2.       Line 118: murine instead of mice

3.       Line 121 Better:… it attracted…

4.       In Figure 3: cleaved instead of clived

The paper would really benefit from another round of thorough proof-reading and minor editing of the English.

Comments on the Quality of English Language

The paper would really benefit from another round of thorough proof-reading and minor editing of the English.

Author Response

Let us first of all thank you for your constructive comments on the review sent to Cells and entitled ‘ TRAIL-non apoptotic signalling”, which the authors appreciated. We are now submitting our revised version, and have addressed all of your comments, as you will see below, with our point-by point answers.

Referee N#1

This is an excellent and comprehensive review of non-canonical TRAIL-receptor signalling. It has a number of informative figures and in my view the article will be of great interest to the scientific community. 

Answer  : Many thanks for highlighting the work done.

Overall, there are a number of typos in the manuscript. A few examples are shown below.

  1. The unfolded-protein response and the involvement of TRAIL-receptor needs to be edited between lines 199 and 211. It is not very clear and hard to follow at the moment.

Answer  : We have rephrased this chapter and hope that it reads better now.

  1. Line 118: murine instead of mice

Answer  : Sorry for this mistake, we have changed the word accordingly

  1. Line 121 Better:… it attracted…

Answer : change made thank you for your proposal.

  1. In Figure 3: cleaved instead of clived

Answer  : Sorry for this mistake, we have changed the word in the figure

The paper would really benefit from another round of thorough proof-reading and minor editing of the English.

Answer  : Thanks for your  advice, we have indeed gone through the manuscript again to correct the few mistakes lefts. All corrections appear in red.

The corrected manuscript is attached 

Please note that we have also changed figure 2 and hope that it reads better. Once again, many thanks for your feedbacks which have helped us produced a review of better quality.

Reviewer 2 Report

Comments and Suggestions for Authors

General comment: The current review article submitted by Guerrache et al is aimed to provide a report on the non-canonical signaling pathways of TRAIL.  The overall manuscript suffers from various flaws including descriptive writing, an unfocused introduction and lack of proper concise discussion. These shortcomings warrant considerable revisions on part of the authors for the manuscript to be deemed publication-worthy.

Comments:

1.      The manuscript is highly descriptive and can be shortened to focus only the major things.

2.      The introduction part does not clearly mention the topics discussed in the paper, however some part of it can be discussed in separate paragraphs. Lines 81-97 need to be discussed as a separate section.

3.      The section “TRAIL-induced cell death” is descriptive, hence needs to be divided into small paragraphs with subheadings.

4.      Line 171-194: Please discuss under a specific subheading.

5.      As the title suggests the non-canonical pathways of TRAIL will be discussed, a major section of the manuscript is highly describing the canonical TRAIL signaling, that needs to be shortened significantly.

6. The figures are highly descriptive with a lot of information. It would be better to focus only on the major points for the betterment of readers’ understanding.

Author Response

Let us first of all thank you for your constructive comments on the review sent to Cells and entitled ‘ TRAIL-non apoptotic signalling”, which the authors appreciated. We are now submitting our revised version, and have addressed all of your comments, as you will see below, with our point-by point answers.

General comment: The current review article submitted by Guerrache et al is aimed to provide a report on the non-canonical signaling pathways of TRAIL.  The overall manuscript suffers from various flaws including descriptive writing, an unfocused introduction and lack of proper concise discussion. These shortcomings warrant considerable revisions on part of the authors for the manuscript to be deemed publication-worthy. 

Answer  : Many thanks for your comments. We have indeed taken your comments into consideration, by shortening as recommended the introduction and removed one figure.

Comments:

  1. The manuscript is highly descriptive and can be shortened to focus only the major things.

Answer  : Dear referee, our aim is to provide a comprehensive review of TRAIL's non-apoptotic signaling. We wished to avoid being dogmatic, reason why we tried to be as exhaustive and fair as possible with the description of the molecular signalling partners described so far to account for TRAIL pleiotropic signalling activities.  We beleive also that the paragraphs dedicated to TRAIL’s physio-pathological functions and comparison with other TNFRSF are necessary to illustrate the importance of these ill-defined signalling pathways We thus think that every single paragraph is important in our review.

  1. The introduction part does not clearly mention the topics discussed in the paper, however some part of it can be discussed in separate paragraphs. Lines 81-97 need to be discussed as a separate section.

Answer  : Thanks for your constructive comment. We have shortened the introduction to focus on the topic, but have kept and rephrased lines 81-90, as we believe that their content serves our purpose.

We have, however placed lines 91 to 97, in the paragraph 3.2.

  1. The section “TRAIL-induced cell death” is descriptive, hence needs to be divided into small paragraphs with subheadings.

Answer  : Thanks, subheadings was added as suggested.

  1. Line 171-194: Please discuss under a specific subheading.

Answer  : Thanks, the following subheading were added for this paragraph. : 1.3 TRAIL-induced necroptosis

  1. As the title suggests the non-canonical pathways of TRAIL will be discussed, a major section of the manuscript is highly describing the canonical TRAIL signaling, that needs to be shortened significantly.

Answer  : As mentioned above, this part cannot be reduced further in our opinion, given that these complexes are intrinsically linked to the secondary complexes that we describe, and whose role in the induction of non-canonical signalling remains hypothetical, unlike the primary complex.

  1. The figures are highly descriptive with a lot of information. It would be better to focus only on the major points for the betterment of readers’ understanding.

Answer  : Thank you for your comment. We agree that Figures 5 and 6 may appear redundant and we have therefore merged them to better explain our intention, in particular to highlight the complexity of these signalling pathways and our lack of understanding of the molecular events underlying the tumour signalling properties of TRAIL. The new Figure 5 is in fact a very good illustration, in our view, that Src/ RAC1 kinases are likely to play an important role in this process. However, and despite the fact that they may be essential in the process, they do not necessarily provide a simple and robust explanation of how these kinases are activated from the TRAIL receptors, and this is indeed nicely illustrated in this new figure 5.

For both referee :

Please note that we have also changed figure 2 and hope that it reads better. Once again, many thanks for your feedbacks which have helped us produced a review of better quality.

Please see attached the corrected version of the manuscript (corrections highlighted in red)
